# Characterization of Proteome Changes in Aged and Collagen VI-Deficient Human Pericyte Cultures

**DOI:** 10.3390/ijms25137118

**Published:** 2024-06-28

**Authors:** Manuela Moriggi, Enrica Torretta, Matilde Cescon, Loris Russo, Ilaria Gregorio, Paola Braghetta, Patrizia Sabatelli, Cesare Faldini, Luciano Merlini, Cesare Gargioli, Paolo Bonaldo, Cecilia Gelfi, Daniele Capitanio

**Affiliations:** 1Department of Biomedical Sciences for Health, University of Milano, 20133 Milano, Italy; manuela.moriggi@unimi.it (M.M.); cecilia.gelfi@unimi.it (C.G.); 2Laboratory of Proteomics and Lipidomics, IRCCS Orthopedic Institute Galeazzi, 20161 Milano, Italy; enrica.torretta@grupposandonato.it; 3Department of Molecular Medicine, University of Padova, 35121 Padova, Italy; matilde.cescon@unipd.it (M.C.); loris.russo@phd.unipd.it (L.R.); ilaria.gregorio1@gmail.com (I.G.); paola.braghetta@unipd.it (P.B.); bonaldo@bio.unipd.it (P.B.); 4CNR-Institute of Molecular Genetics, 40136 Bologna, Italy; sabatelli@area.bo.cnr.it; 5IRCCS Istituto Ortopedico Rizzoli, 40136 Bologna, Italy; 61st Orthopedics and Traumatology Department, IRCCS Istituto Ortopedico Rizzoli, 40136 Bologna, Italy; cesare.faldini@unibo.it; 7Department of Biomedical and Neuromotor Science, DIBINEM, University of Bologna, 40136 Bologna, Italy; luciano.merlini@unibo.it; 8Department of Biology, University of Rome Tor Vergata, 00133 Rome, Italy; cesare.gargioli@uniroma2.it

**Keywords:** pericyte, skeletal muscle, myogenic differentiation, muscle aging, Ullrich congenital muscular dystrophy, Bethlem myopathy

## Abstract

Pericytes are a distinct type of cells interacting with endothelial cells in blood vessels and contributing to endothelial barrier integrity. Furthermore, pericytes show mesenchymal stem cell properties. Muscle-derived pericytes can demonstrate both angiogenic and myogenic capabilities. It is well known that regenerative abilities and muscle stem cell potential decline during aging, leading to sarcopenia. Therefore, this study aimed to investigate the potential of pericytes in supporting muscle differentiation and angiogenesis in elderly individuals and in patients affected by Ullrich congenital muscular dystrophy or by Bethlem myopathy, two inherited conditions caused by mutations in collagen VI genes and sharing similarities with the progressive skeletal muscle changes observed during aging. The study characterized pericytes from different age groups and from individuals with collagen VI deficiency by mass spectrometry-based proteomic and bioinformatic analyses. The findings revealed that aged pericytes display metabolic changes comparable to those seen in aging skeletal muscle, as well as a decline in their stem potential, reduced protein synthesis, and alterations in focal adhesion and contractility, pointing to a decrease in their ability to form blood vessels. Strikingly, pericytes from young patients with collagen VI deficiency showed similar characteristics to aged pericytes, but were found to still handle oxidative stress effectively together with an enhanced angiogenic capacity.

## 1. Introduction

Pericytes are mural cells that surround endothelial cells (ECs) in capillaries, pre-capillary arterioles, and post-capillary venules [1]. They play a critical role in vascular morphogenesis and function by promoting ECs migration, proliferation, and differentiation [2,3]. Pericytes are connected to ECs through intercellular junctions and adhesion plates, involving integrins that link the basement membrane to the plasma membrane, thereby maintaining the endothelial barrier [4]. Additionally, paracrine signaling between pericytes and ECs, mediated by transforming growth factor-beta (TGF-β) and angiopoietin-1 (Ang-1), further supports microvascular barrier function by stabilizing actin filaments in ECs [5,6].

Beyond their role in angiogenesis and endothelial barrier integrity, pericytes also contribute to immune regulation. They can uptake small molecules through phagocytosis, pinocytosis, and endocytosis and are highly responsive to inflammatory signals, guiding leukocytes to inflammation sites [7,8]. Stimulated pericytes secrete pro- and anti-inflammatory molecules, cytokines, and chemokines, such as sphingosine-1-phosphate (S1P), VEGF, and BDNF, which enhance the endothelial barrier function [9,10].

In addition to these functions, pericytes also possess stem cell potential similar to mesenchymal stem cells [11]. They can differentiate into skeletal muscle when transplanted into dystrophic mice, contributing to the smooth muscle layer of blood vessels and the development of skeletal muscle fibers during postnatal growth [12]. Pericytes, which are phenotypically distinct from satellite cells but possess similar myogenic potential, are promising for cell therapy in muscular dystrophies due to their ability to differentiate into both blood vessels and muscle tissue.

It is well known that the process of aging has a negative impact on muscle regeneration and on the potential of myogenic stem cells, leading to muscle tissue depletion or sarcopenia [13,14]. Studies have shown that aged muscle satellite cells have functional alterations such as a decreased ability to form colonies in vitro and a reduced ability to respond to damage [15,16,17,18,19]. Aging also affects the ability of muscle satellite cells to differentiate and replenish reserve pools in culture [20,21,22]. Among the cellular processes controlling myogenic cells’ ability to promote regeneration, autophagy was described to exert a specific role in preserving muscle satellite cell quiescence and regenerative capacity, while it declines with aging, contributing to the induction of senescence [23,24].

Ullrich congenital muscular dystrophy (UCMD) and Bethlem myopathy (BM) are two allelic disorders caused by mutations in genes encoding the different alpha chains of the extracellular matrix (ECM) collagen VI. UCMD and BM display clinical phenotypes ranging from severe (UCMD) to moderate (BM) according to the impact of the defective quantity or altered properties of the collagen VI deposited in the muscular connective tissues [25,26]. In skeletal muscles, the absence or alteration of collagen VI impacts several types of cells’ homeostasis and function, including myoblasts and myofibers, regenerative satellite cells, fibroblasts, and tenocytes [27,28] by modulating different cell processes, such as mitochondrial function and apoptosis, ECM production, cell adhesion, migration, and differentiation, as well as autophagy [29,30]. Moreover, collagen VI defects recapitulate many of the alterations seen in skeletal muscle during the aging process [31].

The present study, for the first time, addresses the potential of skeletal muscle-resident pericytes to support muscle differentiation and angiogenesis in aged subjects and in patients affected by collagen VI-related pathologies. Pericyte cultures from healthy young and aged subjects and young age-matched UCMD and BM patients were investigated by mass spectrometry (MS)-based proteomic analysis with label-free quantification followed by an Ingenuity Pathway Analysis (IPA) bioinformatic approach to highlight similarities and differences in the involvement of specific metabolic pathways or cellular functions in age-related sarcopenia and the pathological condition of collagen VI deficiency.

## 2. Results

We investigated protein changes in primary cell culture extracts of pericytes obtained from a 73-year-old subject (aged pericytes) vs. a 17-year-old young control subject (control pericytes). These results were analyzed together with protein changes obtained from pericyte cultures of 20-year-old patients affected by either UCMD or BM compared to control pericyte cultures. Liquid chromatography coupled to electrospray tandem mass spectrometry (LC–ESI–MS/MS) with label-free quantification was adopted. Out of 1501 identified proteins, the ANOVA test (*p*-value < 0.05) revealed 449 significantly changed proteins in aged compared to control pericytes. Furthermore, 436 proteins were changed in BM and 438 in UCMD pericytes compared to controls (Figure 1). Of these, 170 were in common among aged, BM, and UCMD pericytes vs. control pericytes, whereas 22 and 38 were in common between aged and BM or UCMD pericytes, respectively. On the other hand, 219 proteins were uniquely changed in aged pericytes. BM and UCMD samples shared 176 proteins, whereas 68 characterized the BM and 54 the UCMD pericytes vs. the control. Identification data from LC–ESI–MS/MS of changed proteins are shown in Appendix A (aged vs. control) and Appendix A (UCMD and BM vs. control).

### 2.1. Ingenuity Pathway Analysis: Canonical Pathways

By processing the proteomics datasets using IPA software (winter release 2023), the identification of canonical pathways of aged, BM, and UCMD pericytes cultures compared to healthy young controls was achieved (Table 1). The canonical pathway analysis enabled the recognition of key signaling pathways associated with differentially expressed proteins.

The analysis highlighted both similarities and differences among aged, BM, and UCMD pericytes in terms of signaling pathways. In aged pericytes, carbohydrate metabolism was characterized by a predicted inhibition of glycolysis and gluconeogenesis, while fatty acid β-oxidation and oxidative phosphorylation were predicted to be activated. At variance with this, in BM, carbohydrate metabolism (gluconeogenesis and, although only to a lesser extent, glycolysis) was active. UCMD did not show significant changes (Table 1 and Figure 2).

Concerning cytoskeletal activity and signal transduction, the cell junction and cytoskeleton-related Integrin-Linked Kinase (ILK) signaling was inhibited in all the comparisons. Actin cytoskeleton signaling, regulation of actin-based motility by Rho, and Cell Division Cycle 42 (CDC42) signaling were inhibited in aged pericytes together with the integrin signaling, Rho Family GTPases signaling, Ras Homolog Family Member A (RhoA) signaling, and the concurrent activation of the Rho GDP Dissociation Inhibitor (RhoGDI) signaling. Changes in actin cytoskeleton signaling were retained in BM, whereas UCMD did not show further significant changes (Table 1 and Figure 3).

As regards cellular stress, the Nuclear Factor Erythroid 2 (NRF2)-mediated oxidative stress response was active in both aged and BM pericytes, and only tendentially activated in UCMD pericytes compared to the control.

BM and UCMD samples were characterized by the activation of the pentose phosphate pathway together with aryl hydrocarbon receptor signaling and the glutathione-mediated detoxification system (Table 1 and Figure 4).

Immune activity and inflammation were characterized by the inhibition of leukocyte extravasation signaling in both aged and BM pericytes (Figure 5).

Eukaryotic Translation Initiation Factor 2 (EIF2) signaling, which regulates protein synthesis, was significantly inhibited in aged and BM pericytes, and only tendentially in UCMD pericytes.

In aged pericytes, the Coordinated Lysosomal Expression and Regulation (CLEAR) signaling pathway was inhibited, whereas the Bone Morphogenetic Protein (BMP) signaling and the Janus Kinase/Signal Transducers and Activators of Transcription (JAK/STAT) signaling pathways were activated.

Both BM and UCMD activated apelin signaling, whereas UCMD pericytes only showed activation of C-X-C Chemokine Receptor type 4 (CXCR4) signaling (Table 1 and Figure 5).

### 2.2. Ingenuity Pathway Analysis: Diseases and Biofunctions

IPA analysis of diseases and biofunctions highlighted differences in aged and UCMD pericytes (Table 2). In BM pericytes, only the synthesis of the protein pathway was inhibited, reflecting the milder condition of this pathology. Aged pericytes displayed inhibition of several pathways, including the synthesis and metabolism of protein, attachment of cells, binding of connective tissue cells, adhesion of connective tissue cells, cell movement of epithelial cell lines, shape change of embryonic cell lines, and shape change of epithelial cell lines. In UCMD samples, the epithelial–mesenchymal transition pathway was inhibited, and vice versa; pathways involved with angiogenesis and organismal death were activated.

### 2.3. Autophagy Flux Analysis

Considering that the inhibition of CLEAR signaling was evidenced by bioinformatic analysis and taking into account the role of autophagy in cell quiescence and the maintenance of regenerative capacity and its impairment both in aging and in skeletal muscle with collagen VI deficiency, we decided to investigate autophagy flux also in pericytes.

To monitor the autophagic flux, the levels of lipidated microtubule-associated protein 1A/1B-light chain 3 (LC3II, autophagosomal) over the non-lipidated (LC3I) cytosolic form and sequestosome-1 (P62) were assessed by Western blotting (Figure 6A,B). Basal culture conditions and an autophagy-inductive stimulus, such as serum-deprived Hank’s balanced salt solution (HBSS), were tested. When undifferentiated young pericytes (XX17) were cultured in basal conditions, the addition of Chloroquine (CQ), an inhibitor of the autophagosome–lysosome fusion, induced an increase in the LC3II/LC3I ratio and LC3II protein level, pointing to the presence of a basal active autophagic flux. Notably, CQ’s addition to the pro-autophagic HBSS treatment resulted in an even more significant increase in the LC3II/LC3I ratio and LC3II protein level (Figure 6A,C,D). In contrast, in aged pericytes (XX73), CQ treatment was not able to induce a significant increase in the LC3II/I ratio or LC3II protein level, neither in basal conditions nor upon autophagy induction by HBSS treatment (Figure 6B–D). P62 did not show any significant change in both young and aged cultures (Figure 6E).

In BM and UCMD pericytes, both the LC3II/LC3I ratio and P62 levels were lower when compared to age-matched control pericytes, and the decrease was significant in UCMD cultures, while it showed a trend towards decrease in BM cultures (Figure 6F–I).

### 2.4. Analysis of Senescence

Considering the relevance of autophagy in preventing the onset of senescence in myogenic cells [24], we wondered whether signals of induced senescence could be highlighted in aged pericytes by monitoring the P21 marker [32]. Indeed, immunofluorescence analysis revealed a higher incidence of nuclear localization of P21 in aged pericytes, when compared to younger ones (Figure 7), supporting the occurrence of senescence in these cultures.

## 3. Discussion

Our proteomic and bioinformatic analysis of protein extracts from aged, UCMD, and BM pericytes cultures compared to young healthy controls indicated an alteration in carbohydrate metabolism. In aged pericytes, we found a decrement in glycolytic/gluconeogenic protein levels and an increase in fatty acid oxidation and oxidative phosphorylation. This behavior can be superimposed on what was found during the aging of skeletal muscle, both in humans and in animal models [33,34,35]. In pericytes, the increase in respiration and mitochondrial activities and the production of fatty acids were associated with the loss of cell proliferation, contraction, and extracellular matrix deposition that characterize the naive pericytes and the establishment of a differentiated state [36].

Our data indicate that actin cytoskeleton signaling and RhoA-based motility are strongly inhibited in aged pericytes, pointing at an inhibition of pro-angiogenic signaling. The contractile phenotype of pericytes is dependent on the RhoA/Rho kinase (ROCK) pathway that specifically affects smooth muscle actin-based cytoskeletal networks [37]. Previous studies demonstrated that the enhanced contractility of pericytes has consequences for the proliferation and pro-angiogenic capacity of ECs and that the uncontrolled activation of RhoA signaling leads to increased contractile activity and the disorganization of the pericyte cytoskeleton into stress fibers, resulting in deregulation of the cell cycle and pro-angiogenic behavior of ECs [38]. Integrin signaling was similarly inhibited in aged pericytes, further indicating a disruption of pericyte–EC–extracellular matrix interaction that can lead to impaired vasculogenesis [39,40].

At variance with this result, the NRF2-mediated oxidative stress response was active in aged pericytes. NRF2 plays an essential role in the regulation of redox homeostasis and protection of endothelial cells. The activation of NRF2 in pericytes could be an attempt to preserve the endothelial barrier integrity and help prevent inflammation [41]. In support of this hypothesis, the leukocyte extravasation signaling was inhibited in aged pericytes. Leukocyte extravasation refers to the process by which leukocytes migrate from the bloodstream into tissues during an inflammatory response. Emerging evidence suggests that pericytes are crucial regulators of this process by mediating the crawling of leukocytes across the basement membrane, followed by their adhesion and crawling on pericytes and their migration through pericyte gaps in the interstitial space to the site of inflammation [42].

EIF2 signaling and the CLEAR pathway, involved in the regulation of lysosome-associated processes, such as autophagy, exo- and endocytosis, phagocytosis, and immune response [43], were inhibited as well in aged pericytes, indicating an impairment of protein synthesis and degradation, recycling, and the maintenance of cellular homeostasis.

Since impaired autophagy is a characteristic feature of muscle during aging [44], the autophagy flux was investigated in more detail in the different pericyte cultures by immunoblotting for LC3 lipidation. The obtained results indicate an impairment of the autophagic flux in aged pericytes, evidencing a loss of angiogenic activity and function [45]. Aging-related autophagy impairment was previously indicated as a mechanism underlying a decline in the regenerative potential of other myogenic cells [23,46].

On the contrary, BMP and JAK/STAT pathways were activated. BMPs are signaling molecules belonging to the superfamily of TGF-β and play an important role in embryogenesis and development as well as in the maintenance of the homeostasis of adult tissues [47]. BMP signals are capable of regulating endothelial cell adhesion junctions via either a pro-angiogenic or anti-angiogenic signaling mechanism, respectively resulting in the activation or stabilization of the junctions [48]. The JAK/STAT pathway is the principal signaling mechanism for a wide array of cytokines and growth factors, resulting in cell proliferation, differentiation, migration, and apoptosis [49]. The activation of this pathway has been correlated with pericyte-induced aggravation of a lowered ECs’ barrier functioning in rat brains [50].

In agreement with the above results, the analysis regarding diseases and biofunctions predicted the inhibition of protein synthesis and metabolism, together with the inhibition of cell adhesion, movement, and shape change in aged pericytes.

Pericytes obtained from BM and UCMD patients were characterized by an inhibition of ILK signaling compared to healthy age-matched controls, similar to what we observed in aged pericytes. Furthermore, BM pericytes displayed some additional features in common with aged pericytes (i.e., actin cytoskeleton signaling, NRF2-mediated oxidative stress response, leukocyte extravasation signaling, and EIF2 signaling).

At variance with aged pericytes, both BM and UCMD pericytes showed the activation of the pentose phosphate pathway together with the glutathione-mediated detoxification system, indicating an increased capacity to respond to oxidative stress and inflammation. Furthermore, BM and UCMD pericytes activated aryl hydrocarbon receptor signaling, which has been found to serve an important role in immune responses and during inflammation, oxidative stress, and lipid deposition [51]. Both BM and UCMD pericytes also showed the activation of apelin signaling, which plays important and various roles in cardiac and vascular development; it stimulates angiogenesis and cell migration while reducing vascular tone [52,53] and is associated with skeletal muscle regeneration after injury [54].

UCMD pericytes were further characterized by the activation of CXCR4 signaling, a factor controlling pericyte recruitment, proliferation, and pericyte-induced basement membrane deposition during capillary network assembly [55], suggesting an enhanced angiogenic response, which was confirmed also by the diseases and biofunction analysis, together with the inhibition of the epithelial–mesenchymal transition and the activation of the organismal death process, which indicate that cells lose their mesenchymal features and activated pro-apoptotic signaling.

Immunoblotting for autophagic markers evidenced a decrease in LC3 lipidation and P62 levels, which were statistically significant in UCMD vs. control pericytes. This appears in line with the idea that a defect in collagen VI can modulate the autophagic process. This was highlighted by an analysis performed in tissues from the collagen VI knockout mouse [56,57] as well as in cells and tissues derived from BM/UCMD patients [57,58,59]. While the autophagic defect was so relevant for collagen VI studies that it became a major druggable target for preclinical therapeutic strategies [57,60,61] as well as for pilot clinical trials [58], the evidence of an autophagy defect in pericytes was never investigated so far.

Taken together, these data indicate that aged pericytes undergo metabolic alterations, which reflect those observed in aged skeletal muscle and are correlated to the loss of stem cell potential. Moreover, our proteomics analysis indicates that aged pericytes are characterized by a decreased protein synthesis and by an impaired focal adhesion and contractility that, together with the inhibition of the integrin signaling, are indicative of a loss of vasculogenic/angiogenic function. The features exhibited by UCMD and BM pericytes appear to anticipate those observed in aged pericytes. However, in contrast to the latter, they maintain the capacity to cope with oxidative stress, and, particularly in the case of UCMD, angiogenic power is preserved, albeit at the expense of stemness potential.

The main limitation of this study lies in the small number of subjects involved. Further studies involving more subjects showing similar characteristics for each group will be considered for future developments in order to validate the results and increase their confidence. Nevertheless, our results, while highlighting the potential of pericytes in muscle development, emphasize that the effective use of these cell types in muscle regeneration therapy requires treatments aimed to rejuvenate their condition by improving their crosstalk with their environment and their surrounding cell types.

## 4. Materials and Methods

### 4.1. Cell Cultures and Treatments

Aged and control pericytes were isolated from human skeletal muscle biopsies provided by Prof. Biagini at the “Istituto Regina Elena (IRE)” cancer hospital in Rome. Skeletal muscle biopsies from 2 genetically characterized patients with COLVI-related myopathies were collected at the Istituto Ortopedico Rizzoli and CNR of Bologna. The patient affected by BM carried a COL6A2 exon 6 c.802 G>A, p.Gly268Ser heterozygous mutation (P11 in Merlini et al., 2023 [62]), and the UCMD patient an exon 10 c.896 G>A, p.Gly299Glu heterozygous mutation in COL6A1 gene (P40 in Merlini et al., 2023 [62]).

Biopsies were transported from the operating room to the laboratory in refrigerated saline; the tissue samples were immediately processed under a sterile laminar flow hood. We selected pericytes by their capacity to attach to plastic (without coating) and grow in this relatively poor condition at low cell confluence (1 × 10^3^ cell/cm^2^) according to previously published protocols [63]. Pericytes were selected by plastic adherence in culture for at least 10 days when they formed colonies positive for alkaline phosphatase (ALP), neural/glial antigen 2 (NG2), and CD146 [64]. Briefly, the tissue was finely minced and collected in a solution, 100 U/mL of type II collagenase (GIBCO, Life Technologies, Carlsbad, CA, USA). The tube was placed in a thermal shaker at 37 °C for 45 min. At the end of digestion, the resulting solution was centrifuged for 10 min at 1100 rpm and resuspended in α-MEM growth medium (Minimum Essential Medium-Alpha, Thermo Fisher Scientific, Waltham, MA, USA) supplemented with 20% fetal bovine serum (FBS). The cell suspension was filtered through a 70 mm cell strainer, and dispensed in plastic dishes at clonal density (1 × 10^3^ cell/cm^2^). The isolated cells were cultured at 37 °C and 5% CO_2_ in α-MEM, supplemented with penicillin 100 U/mL, streptomycin 100 µm/mL, 1 mM sodium pyruvate (NaPyr), 1 mM HEPES, with the addition of 20% FBS. Culture medium was replaced every 48 h during expansion. After 10–15 days from seeding, ALP-positive colonies started to form; thus, pericytes were expanded and tested for their myogenic and angiogenic potential. Cells were detached by using 0.25% trypsin-EDTA (Thermo Fisher Scientific) at 60–70% confluency for further expansion or plating.

The study of autophagic flux was conducted by changing the medium for one last hour of culture for all the following four conditions: (i) fresh basal growth medium; (ii) fresh basal medium with Chloroquine 50 μM (Merck, Darmstadt, Germany); (iii) Hank’s balanced salt solution (HBSS, Thermo Fisher Scientific); (iv) HBSS with Chloroquine 50 μM (Merck).

### 4.2. Protein Extraction for Label-Free LC–ESI–MS/MS Analysis

Cells were suspended in lysis buffer (2% SDS, 100 mM Tris-HCl pH 7.6, 0.1 M DTT, and 1% phenylmethanesulfonylfluoride) and sonicated on ice until completely dissolved. After incubation at 95 °C for 3 min, lysates were clarified by centrifugation at 12,000× *g* for 20 min at 4 °C. Protein quantitation with 2-D Quant-kit protein assay (Cytiva, Little Chalfont, UK) was then performed.

### 4.3. Label-Free Liquid Chromatography with Tandem Mass Spectrometry

Protein extracts (100 µg for each sample) were processed following the filter-aided sample preparation (FASP) protocol [65]. Peptide samples were concentrated, and separated on a Dionex UltiMate 3000 HPLC System with an Easy Spray PepMap RSLC C18 column (250 mm, internal diameter of 75 µm) (Thermo Fisher Scientific), and electrosprayed into an Orbitrap Fusion Tribrid (Thermo Fisher Scientific) mass spectrometer. Three technical replicates for each sample were acquired. Mass spectra were analyzed using MaxQuant software (Max Planck Institute of Biochemistry, Munich, Germany, version 1.6.17.0) [66]. The maximum allowed mass deviation was set to 6 ppm for monoisotopic precursor ions and 0.5 Da for MS/MS peaks. Enzyme specificity was set to trypsin/P, and a maximum of two missed cleavages was allowed. Carbamidomethylation was set as a fixed modification, while N-terminal acetylation and methionine oxidation were set as variable modifications. Spectra were searched by the Andromeda search engine against the *Homo sapiens* Uniprot UP000005640 sequence database (78.120 proteins, released 7 March 2021). Protein identification required at least one unique or razor peptide per protein group. Quantification in MaxQuant was performed using the built-in extracted ion chromatogram (XIC)-based label-free quantification (LFQ) algorithm using fast LFQ [67]. The required FDR was set to 1% at the peptide, 1% at the protein, and 1% at the site-modification levels, and the minimum required peptide length was set to 7 amino acids. Statistical analyses were performed using Perseus software (v.1.6.15.0, Max Planck Institute of Biochemistry, Martinsried, Germany) [68]. For each experimental group, proteins identified in at least 80% of samples were considered. For statistical analysis, an ANOVA test with a *p*-value threshold of 0.05 was applied. False positives were reduced utilizing the Benjamini–Hochberg false discovery rate test.

### 4.4. Ingenuity Pathway Analysis

Functional and network analyses of statistically significant protein expression changes were performed through Ingenuity Pathway Analysis (IPA, winter release 2023) software (Qiagen, Hilden, Germany). In brief, datasets with protein identifiers, statistical test *p*-values, and fold change values calculated from label-free LC–ESI–MS/MS were analyzed by IPA. The “core analysis” function was used for data interpretation through the analysis of biological processes, canonical pathways, diseases, and biofunctions enriched with proteins differentially regulated. Then, the “comparison analysis” function was used to visualize and identify significant proteins or regulators across experimental conditions. *p*-Values were calculated using a right-tailed Fisher’s exact test. The activation z-score was used to predict the activation/inhibition of a pathway/regulator/disease and biofunction [69]. A Fisher’s exact test *p*-value < 0.05 and z-scores ≤ −2 and ≥2, which takes into account the directionality of the observed effects, were considered statistically significant.

### 4.5. Western Blotting for the Analysis of Autophagic Flux

Protein extracts were obtained from pericytes washed and scraped after adding RIPA lysis buffer containing 50 mM Tris-HCl (Merck), 150 mM NaCl (Merck), 1% IgePal (Merck), 0.5% sodium deoxycholate (Merck), 0.1% SDS (Merck), supplemented with EDTA-free protease inhibitors (Roche, Basel, Switzerland) and phosphatase inhibitor cocktail II (Merck). Samples were centrifuged at 4° C at 16,000× *g* and the protein-containing supernatants were collected. To assess protein concentration, the BCA Protein Kit Assay (Thermo Fisher Scientific) was applied. Protein samples were then prepared and blotted as follows. Briefly, extracts in reducing sample buffer were denatured and then loaded on a polyacrylamide gel (Thermo Fisher Scientific) for SDS-PAGE. Blotted PVDF membranes (Thermo Fisher Scientific) were blocked in 5% non-fat dry milk in 1% Tween-20 (Merck) tris-buffered saline (TBST). Primary antibodies for LC3 (Cell Signaling Technology, Danvers, MA, USA, #2775), P62 (Merck, #P0067), and vinculin (Merck #V4505) were diluted in 5% BSA or 2.5% milk in TBST and incubated overnight at 4 °C. After washing, membranes were incubated with HRP-conjugated secondary antibodies (Cytiva) for 1 h at room temperature. Signals were visualized by chemiluminescence using the ECL Prime detection kit and the Image Quant LAS 4000 (Cytiva) analysis system. Band quantification was performed using Image Quant TL (Cytiva, version v8.1.0.0) software followed by statistical analysis. Band intensities were normalized against the total amount of proteins stained by Sypro Ruby total-protein stain.

### 4.6. Immunofluorescence

Immunofluorescence was performed on cells plated on glass coverslips. On the last day of culture, cells were washed and fixed with 4% PFA in PBS for 10 min. After washing with PBS, cells were permeabilized with a solution of 0.2% Triton X-100 in PBS for 5 min. After washing, cells were blocked in 10% goat serum, then incubated with an anti-P21 primary antibody (Santa Cruz Biotechnology, Dallas, TX, USA, sc-6246) in 5% goat serum in PBS overnight at 4 °C. Cells were washed in PBS and incubated with secondary antibodies and Phalloidin-488 (Thermo Fisher Scientific, A12379) in 5% goat serum in PBS for 1 h at RT. Glass slides were then mounted with 80% glycerol in PBS and images were acquired using confocal (Zeiss LSM700, Leica Stellaris 8) or fluorescent microscopes (Leica DM5000B). Immunofluorescence intensity was quantified on unprocessed original images.

### 4.7. Statistical Analysis

Statistical analysis was performed using GraphPad Prism Software v.8.0.2. The statistical tests used are described in the respective figure legends. If not otherwise specified, an unpaired two-tailed Student’s *t*-test was used to compare differences between genotypes for normally distributed data. *p*-Values equal to or less than 0.05 were considered statistically significant. N corresponds to independent biological replicates.

## Figures and Tables

**Figure 1 ijms-25-07118-f001:**
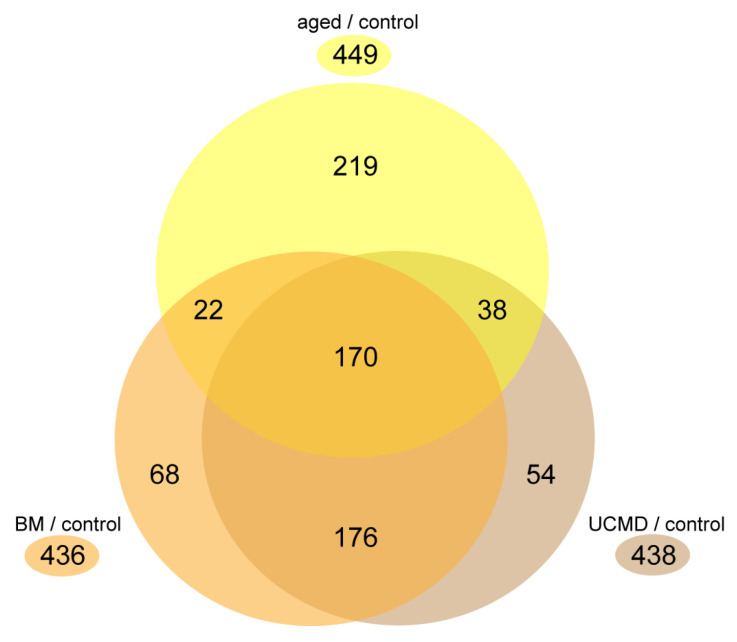
Venn diagram comparing the number of changed proteins in aged (yellow), BM (orange), and UCMD (brown) pericyte cultures compared to controls.

**Figure 2 ijms-25-07118-f002:**
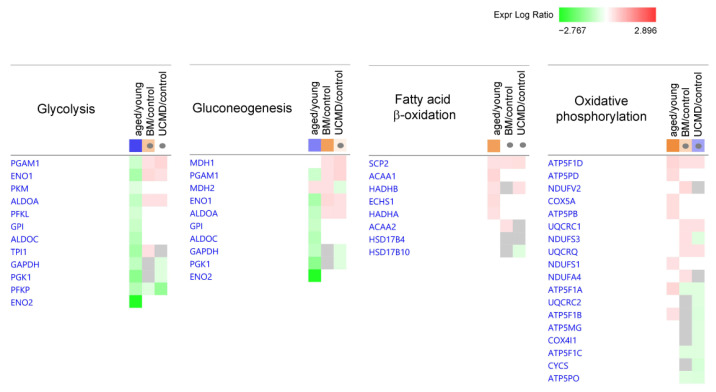
Heatmaps showing dysregulated proteins belonging to the IPA canonical pathways involved in carbohydrate metabolism. Proteins are indicated by gene names; the full names are given in Appendix A. Green and red colors refer, respectively, to a decrease or increase for each protein statistically changed in the obtained proteomics dataset. The grey color indicates changes that are statistically not significant in terms of *p*-value (ANOVA + FDR followed by Tukey’s post hoc test (*p* < 0.05)). Grey dots indicate non-significant values for z-score (significant z-score ≥ 2, ≤−2).

**Figure 3 ijms-25-07118-f003:**
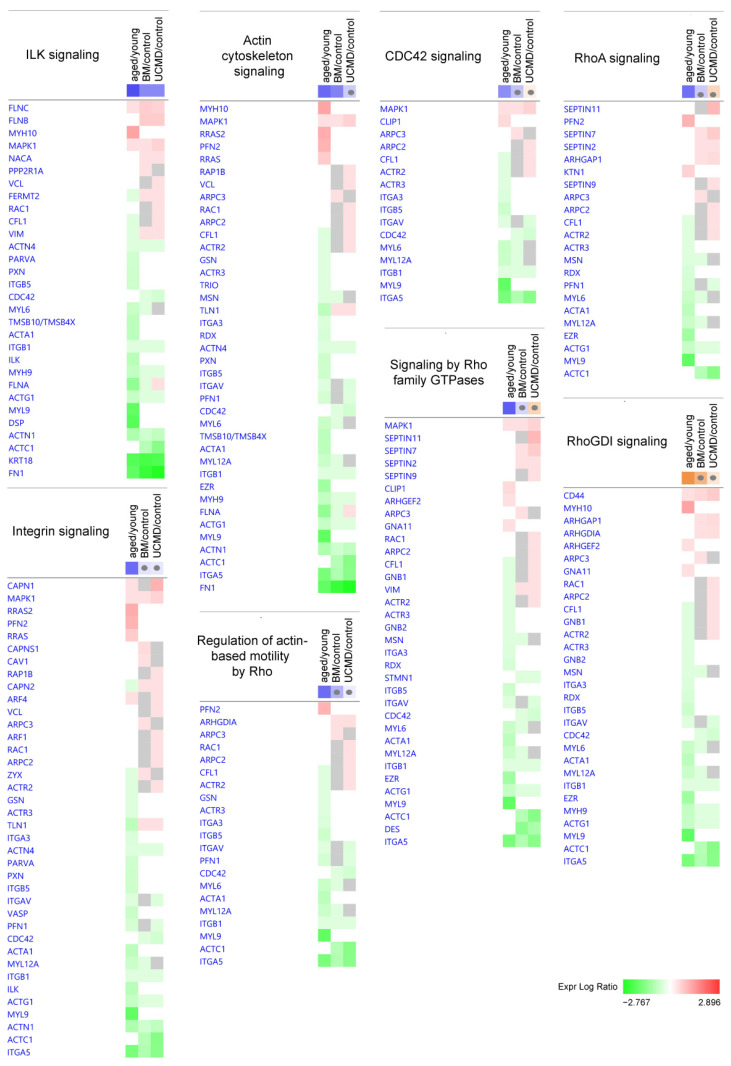
Heatmaps showing dysregulated proteins belonging to the IPA canonical pathways involved in cytoskeletal activity and signal transduction. Proteins are indicated by gene names; the full names are given in Appendix A. Green and red colors refer, respectively, to a decrease or increase for each protein statistically changed in our proteomics dataset. The grey color indicates changes that are statistically not significant in terms of *p*-value (ANOVA + FDR followed by Tukey’s post hoc test (*p* < 0.05)). Grey dots indicate non-significant values for z-score (significant z-score ≥ 2, ≤−2).

**Figure 4 ijms-25-07118-f004:**
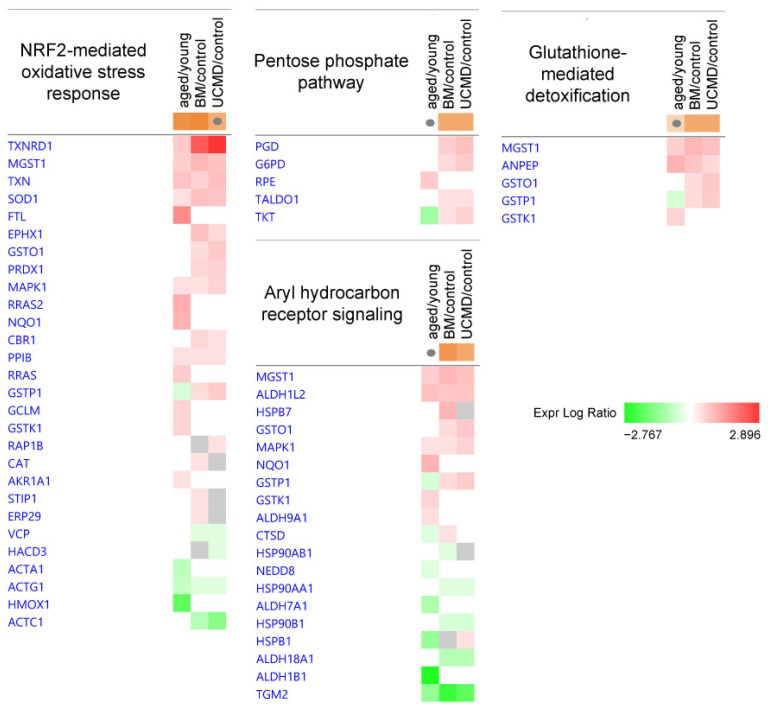
Heatmaps showing dysregulated proteins belonging to the IPA canonical pathways involved in cellular stress. Proteins are indicated by gene names; the full names are given in Appendix A. Green and red colors refer, respectively, to a decrease or increase for each protein statistically changed in our proteomics dataset. The grey color indicates changes that are statistically not significant in terms of *p*-value (ANOVA + FDR followed by Tukey’s post hoc test (*p* < 0.05)). Grey dots indicate non-significant values for z-score (significant z-score ≥ 2, ≤−2).

**Figure 5 ijms-25-07118-f005:**
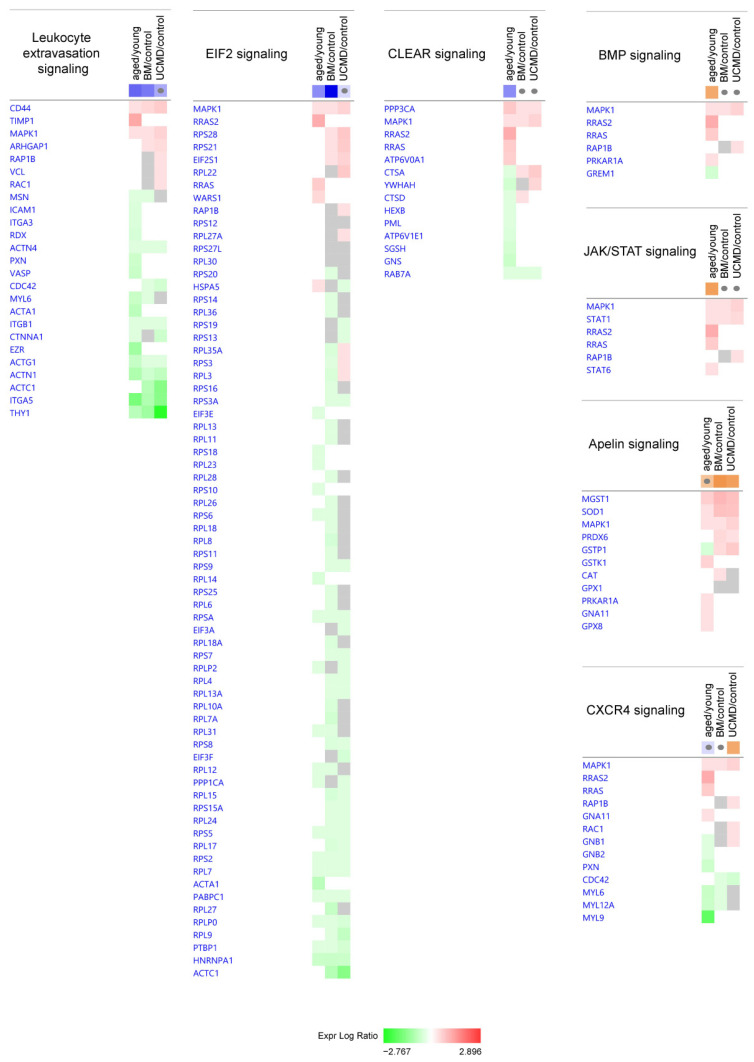
Heatmaps showing dysregulated proteins belonging to the IPA canonical pathways involved in cell signaling. Proteins are indicated by gene names; the full names are given in Appendix A. Green and red colors refer, respectively, to a decrease or increase for each protein statistically changed in our proteomics dataset. The grey color indicates changes that are statistically not significant in terms of *p*-value (ANOVA + FDR followed by Tukey’s post hoc test (*p* < 0.05)). Grey dots indicate non-significant values for z-score (significant z-score ≥ 2, ≤−2).

**Figure 6 ijms-25-07118-f006:**
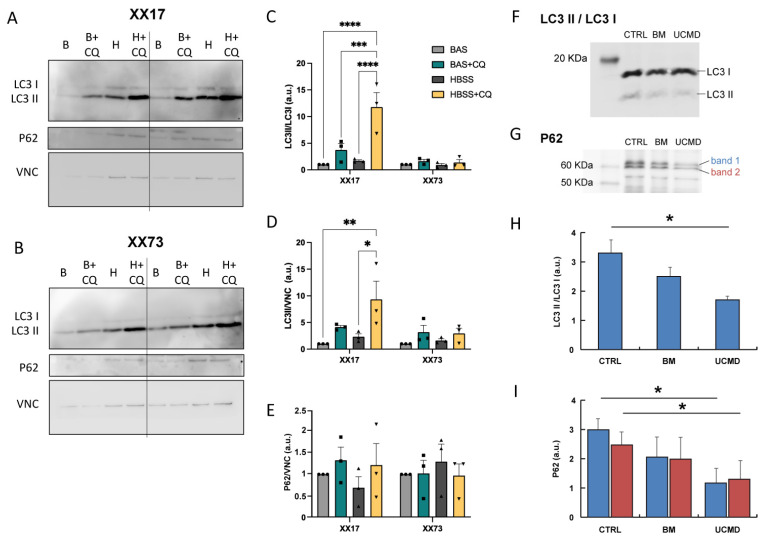
(**A**,**B**) Representative Western blot images of the autophagic markers, LC3 and P62 proteins, on protein extracts from young (XX17, **A**) and aged (XX73, **B**) pericyte cultures. Vinculin (VNC) was used as loading control. Cells were analyzed in different conditions: in basal medium (**B**), upon addition of Chloroquine to basal medium (B + CQ), in HBSS (**H**), and upon addition of Chloroquine to HBSS (H + CQ). (**C**–**E**) Quantification of LC3II/LC3I ratio (**C**), and of LC3II (**D**) or P62 (**E**) protein level, normalized on the level of vinculin (VNC) (*n* = 3 independent experiments; *, *p* < 0.05; **, *p* < 0.01; ***, *p* < 0.001; ****, *p* < 0.0001; 2-way ANOVA with Holm–Šídák’s multiple comparisons test). (**F**,**G**) Representative Western blot images of LC3 and P62 proteins, on protein extracts from control, BM, and UCMD pericyte cultures. (**H**,**I**) Quantification of LC3II/LC3I ratios (**H**) and P62 (**I**) protein levels in control, BM, and UCMD pericyte cultures (ANOVA and Tukey’s multiple comparisons test, *n* = 2, *p* < 0.05). Blue bars = upper molecular weight band (band 1 in panel **G**); red bars = lower molecular weight band (band 2 in panel **G**).

**Figure 7 ijms-25-07118-f007:**
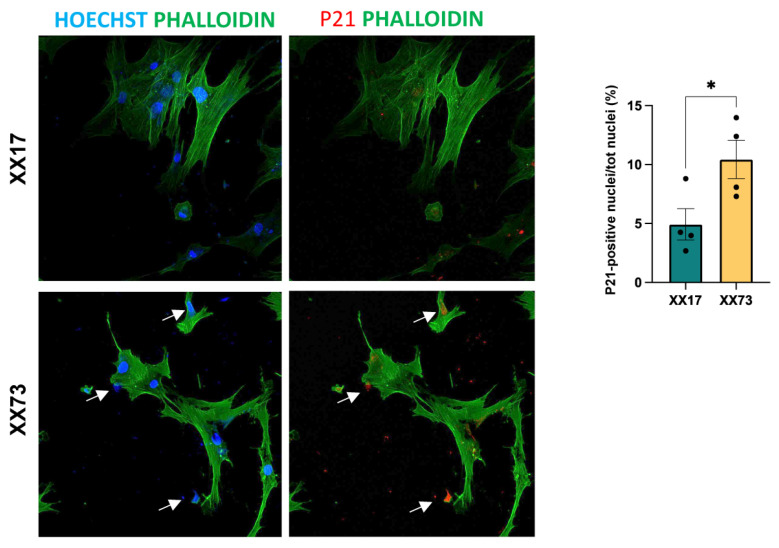
Representative confocal images of immunofluorescence analysis of the senescence marker P21 (red) in young (XX17) and aged (XX73) pericyte cultures. Cell morphology is defined by the staining with Phalloidin-488 (green). Quantification is provided in terms of percentage of P21-positive nuclei (red; white arrows) on the total nuclei (blue, Hoechst) (*n* = 4 experimental replicates; *, *p* < 0.05; unpaired Student’s *t*-test).

**Table 1 ijms-25-07118-t001:** Heatmap of canonical pathways displaying the most significant results (ordered by increasing z-scores in aged vs. control) resulting from IPA analysis. Orange and blue indicate predicted pathway activation or predicted inhibition, respectively, via the z-score statistic (significant z-scores ≥ 2, ≤−2, in bold). N/A: not applicable.

Canonical Pathways	Aged/Control	BM/Control	UCMD/Control
Glycolysis I	**−3.464**	1.342	0
Gluconeogenesis I	**−2.333**	**2.236**	0.378
Fatty Acid β-oxidation I	**2.236**	N/A	N/A
Oxidative Phosphorylation	**2.646**	1	−1.732
ILK Signaling	**−3.273**	**−2.183**	**−2.183**
Signaling by Rho Family GTPases	**−3**	−0.775	1
Actin Cytoskeleton Signaling	**−2.785**	**−2.324**	−0.943
Integrin Signaling	**−2.746**	−0.535	−0.447
Regulation of Actin-based Motility by Rho	**−2.714**	−1.414	−0.333
RHOA Signaling	**−2.673**	−1	0.905
CDC42 Signaling	**−2.111**	−1.134	0.378
RHOGDI Signaling	**2.5**	1.732	0.577
Pentose Phosphate Pathway	N/A	**2**	**2**
Aryl Hydrocarbon Receptor Signaling	0	**2.449**	**2**
Glutathione-mediated Detoxification	1	**2**	**2**
NRF2-mediated Oxidative Stress Response	**2.53**	**2.714**	1.897
Leukocyte Extravasation Signaling	**−2.828**	**−2.496**	−1.604
EIF2 Signaling	**−2.138**	**−4.707**	−0.728
CLEAR Signaling Pathway	**−2.138**	N/A	N/A
CXCR4 Signaling	−0.707	N/A	**2**
Apelin Adipocyte Signaling Pathway	1.414	**2.449**	**2.236**
BMP signaling pathway	**2**	N/A	N/A
JAK/STAT Signaling	**2.236**	N/A	N/A

**Table 2 ijms-25-07118-t002:** Heatmap of diseases and biofunctions displaying the most significant results (ordered by increasing z-scores) resulting from IPA analysis. Orange and blue indicate predicted pathway activation or inhibition, respectively, via the z-score statistic (significant z-scores ≥ 2, ≤−2, in bold). N/A: not applicable.

Diseases and Biofunctions	Aged/Control	BM/Control	UCMD/Control
Synthesis of protein	**−2.875**	**−2.06**	−0.459
Metabolism of protein	**−2.589**	−1.764	−0.797
Attachment of cells	**−3.065**	−0.288	−1.16
Binding of connective tissue cells	**−2.717**	−0.694	−0.548
Adhesion of connective tissue cells	**−2.531**	−0.141	−0.423
Cell movement of epithelial cell lines	**−2.462**	−0.342	0.385
Shape change of embryonic cell lines	**−2.449**	N/A	−0.447
Shape change of epithelial cell lines	**−2.138**	−1.664	−0.896
Epithelial−mesenchymal transition	N/A	−0.116	**−2.029**
Angiogenesis	−1.723	1.421	**2.29**
Organismal death	−1.257	1.069	**2.673**

## Data Availability

The original contributions presented in the study are included in the article/Appendix A; further inquiries can be directed to the corresponding author.

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
