# Peer review of "Characterization of Proteome Changes in Aged and Collagen VI-Deficient Human Pericyte Cultures"

_ijms, 2024, doi:10.3390/ijms25137118_

Round 1

Reviewer 1 Report

Comments and Suggestions for Authors

This study employed mass spectroscope-based proteomic and bioinformatic analyses to characterize periosteal cell collagen deficiency (VI-CD) from different age groups and type VI patients. The study aimed to explore the potential of periosteal cells in muscle differentiation and angiogenesis in elderly and congenital Ulrich patients with muscular dystrophy or Berim myopathy. While this is an interesting research topic, the authors seem to have limited their focus to proteomic analysis without delving into the differences between different proteins and the common proteome. Instead, they turned to verifying autophagy problems in young people and aging periosteal cells. This shift may make the research's themes and goals become reality less prominent. Why did the authors not screen for differential proteins? This could have identified several candidates for protein function and mechanism studies that are closely related to specific molecular phenotypes.

Comments on the Quality of English Language

In terms of manuscript writing, although the overall quality is good, there is still room for improvement. :

1. The introduction may be too long, so it is suggested to simplify this part and remove the background introduction that is not relevant to the research of this paper.

2. Add "Table 1" before "Figure" in L138, 154, L168 and L185, because I think the results in Table 1 can better explain the signaling pathways in the three control groups. The corresponding numbers show only genetic differences on this basis.

3. For uniformity, manuscripts should consistently deal with the capitalization of the "P".

4. As for the references cited in the manuscript, it is found that there are few references in the past five years, so it is suggested that the author try to cite the latest references. Because the research results in recent years are more representative and realistic, they can better support the conclusions and views of the current research.

Author Response

The authors thank the reviewer for suggestions aimed at improving the quality of the article. In particular, the following changes have been included:

  1. Following the Reviewer's suggestions, the introduction has been shortened and simplified.
  2. The reference to "Table 1" has been inserted where suggested.
  3. Capitalization of the letter “P” has been standardized throughout the manuscript.
  4. Six new, more up-to-date references (2021-2024) have been included in the text (3 in Introduction and 3 in Discussion sections).

Reviewer 2 Report

Comments and Suggestions for Authors

Pericytes surround endothelial cells of capillaries and are essential for maintaining the vasculature and performing immune functions such as phagocytosis. Interestingly, pericytes also exhibit characteristics of mesenchymal stem cells, including the ability to differentiate into various cell types. Aging diminishes the regenerative capacity of multiple stem cell pools, and in the present study, the authors sought to define the ability of pericytes from skeletal muscle to support muscle differentiation and angiogenesis in normal aging and collagen IV-related pathologies. The results are interesting, and with some edits, the manuscript could be improved:

1. In Figure 1, the description points to the comparison of cells isolated from one 17-year-old versus those of one 73-year-old. The experiment probably should have been repeated with at least two additional subjects from each group to increase confidence in the reported results. At a minimum, discuss this limitation in the manuscript.

2. In the Methods section, the procedure for isolating cells does not seem to be specific for pericytes, instead culturing a heterogeneous group of cell types. Please elaborate on the steps taken to isolate pericytes.

3. If Figure 6, is this also focused on cells from two individuals: one who is 17 and one who is 73? These experiments could have been improved by using cells isolated from multiple individuals representing similar characteristics. At a minimum, note this as a limitation as well. 

4. The evidence presented in Figure 7 would have been much stronger with additional senescence markers, such as P16Ink4a.

Author Response

The authors thank and totally agree with the reviewer's suggestions aimed at improving the quality of the manuscript and hope that the changes introduced will meet the reviewer's demands.

Reviewer’s points 1 and 3: a sentence was introduced at the end of the discussion highlighting the limitations of the study, as correctly suggested by the reviewer.

Reviewer’s point 2: Pericytes isolation section has been rewritten by better detailing, and introducing a new reference (ref. 64).

Reviewer’s point 4: the authors agree with the reviewer's opinion. Unfortunately, it was not possible to increase the number of senescence markers due to the limited availability of samples. We will consider assaying additional markers for subsequent analyses.

Round 2

Reviewer 2 Report

Comments and Suggestions for Authors

Issues have been addressed.